# Investigations on Mechanical Properties of Lattice Structures with Different Values of Relative Density Made from 316L by Selective Laser Melting (SLM)

**DOI:** 10.3390/ma13092204

**Published:** 2020-05-11

**Authors:** Paweł Płatek, Judyta Sienkiewicz, Jacek Janiszewski, Fengchun Jiang

**Affiliations:** 1Faculty of Mechatronics and Aerospace, Military University of Technology, 2 Gen. S. Kaliskiego Street, 00-908 Warsaw, Poland; judyta.sienkiewicz@wat.edu.pl (J.S.); jacek.janiszewski@wat.edu.pl (J.J.); 2College of Material Science and Chemical Engineering, Harbin Engineering University, 145 Nan-Tong Street, Harbin 15000, China; fengchunjiang@hrbeu.edu.cn

**Keywords:** lattice structures, additive manufacturing, selective laser melting, powder bed fusion, energy absorption, dynamic compression, crashworthiness

## Abstract

Nine variants of regular lattice structures with different relative densities have been designed and successfully manufactured. The produced structures have been subjected to geometrical quality control, and the manufacturability of the implemented selective laser melting (SLM) technique has been assessed. It was found that the dimensions of the produced lattice struts differ from those of the designed struts. These deviations depend on the strut orientation in relation to the specimen-building direction. Additionally, the microstructures and phase compositions of the obtained structures were characterized and compared with those of conventionally produced 316L stainless steel. The microstructure analysis and X-ray diffraction (XRD) patterns revealed a single austenite phase in the SLM samples. Both a certain broadening and a displacement of the austenite peaks were observed due to residual stresses and a crystallographic texture induced by the SLM process. Furthermore, the mechanical behavior of the lattice structure material has been defined. It was demonstrated that under both quasi-static and dynamic testing, lattice structures with high relative densities are stretch-dominated, whereas those with low relative densities are bending-dominated. Moreover, the linear dependency between the value of energy absorption and relative density under dynamic loading conditions has been established.

## 1. Introduction

Over the last few decades, a growing influence of additive manufacturing (AM) methods of production in industrial applications has been observed [1,2,3,4,5,6]. Initially, these methods were generally used in rapid prototyping processes to develop new products [3]. Thanks to AM, it was possible to shorten the duration of the design process and evaluate adopted design assumptions in a rapid and easy manner. During this period of time, additive techniques such as stereolithography (SLA), fused deposition modeling (FDM), 3D printing, and laminated object manufacturing (LOM) were generally used [7,8]. Current developments in this field have created growing interest in metal additive manufacturing (MAM) techniques [9,10], which have opened new perspectives for leading branches of industry as well as medicine and art. Commonly available powder bed fusion (PBF) and direct energy deposition (DED) systems enable the crafting of objects made from a wide variety of metal powders with mechanical properties similar to those of bulk materials. Thanks to the “layer-by-layer” method of manufacturing, devices using PBF (such as selective laser melting (SLM), direct metal laser sintering (DMLS), and electron beam melting (EBM)) allow design freedom and for building parts with complex shapes and geometries that are not achievable using typical subtractive methods [9,11]. DED devices such as laser-engineered net shaping (LENS) [12,13,14,15] and wire arc additive manufacturing (WAAM) [16,17] enable the building of objects with gradient mechanical properties that are defined by the type and amount of applied powder materials or metal wires during the manufacturing process.

Based on data presented in scientific reports, it can be observed that the aforementioned metal additive manufacturing methods are commonly used in aviation and automotive industrial applications [1,18]. They are applied to build “ready-to-use” mechanical parts as well as tools (molds, dies, punches, etc.) [19,20]. Over the past few years, MAM methods have been used in the production of regular cellular structures [21,22,23,24] that demonstrate high energy absorption capacities with low relative densities [25,26,27,28]. In addition, these materials often exhibit other interesting properties that make them attractive as construction materials (thermal, sound and vibration insulation, among others) [23,29,30,31,32]. Moreover, they can be used in new cutting-edge passive protective systems dedicated to both military [33,34,35,36,37,38] and civilian applications. Depending on the applied method of manufacturing (DED or PBF), they are built as 2D (cellular) or 3D (lattice) structures. The first group is generally defined based on 2D topology designed with the use of various unit cell shapes and sizes, which are extruded in the third dimension. One of the well-known representatives of this material group is the honeycomb, which has been investigated in many scientific papers [39,40,41,42,43,44]. This group of structures requires different deformation processes depending on the applied loading orientation (in-plane or out-of-plane) [24,45,46,47,48,49,50,51]. The other group of regular cellular structures is represented by 3D lattices [52,53,54,55,56]. They are characterized by lower values of relative density compared to those of 2D structures, but the possibilities of the structure topology design are more limited due to the three-dimensional shape of the unit cell. The low mass of 3D lattice structures has led many scientists to study their mechanical properties. Ozdemir et al. [57] specified the behavior of 3D-type structures made from Ti-6Al-4V with different unit cell topologies under compression tests. Based on the results obtained, they found that lattice structures effectively spread a peak force during dynamic loading conditions over a short period of time. Leary et al. [52] studied the influence of the lattice topology on the deformation process plot. They also analyzed the manufacturability of the lattice structures depending on the adopted additional inclinations and struts in the unit cell geometry. Lijun and Weidong [58] evaluated the deformation plot of graded lattice structures made from Ti-6Al-4V titanium alloy under various loading conditions and found a strong relationship between the applied topology of the lattice structure and its capacity for energy absorption. Another work by Choy et al. [59] analyzed the influence of titanium alloy lattice structure mechanical properties according to their topology and orientation defined during the manufacturing process. They revealed that structures with higher relative densities during compression collapse exhibited more rupturing due to their high stiffness. Moreover, they observed that the collapse of the structure is caused by the stretch-dominated damage mechanism, rather than by bending. The results presented by Harris et al. [60] suggest that lattice structures have a high potential for optimization. They compared the mechanical responses of additively manufactured SS316L stainless steel hybrid lattices to those of honeycomb structures under quasi-static and dynamic loading conditions. Their results also showed that in terms of the energy absorption efficiency, hybrid lattices demonstrate the highest performance and could be optimized.

All of the aforementioned works were focused on searching for new ultra-light solutions with high effectiveness in terms of energy absorption. Based on the presented results, it can be claimed that lattice structures are one of the most likely groups of materials to fulfill these requirements.

The main goal of the paper is a presentation of results of experimental investigations related to the mechanical properties of lattice structures made additively from SS316L stainless steel. Developed specimens with different values of relative density were manufactured via selective laser sintering SLM and submitted to mechanical and structural tests according to methodology presented in Figure 1. This paper is a continuation of the authors’ works related to additively manufactured lattice structure made from 316L stainless steel. Results associated with this work were partially published in [61,62,63].

On the basis of results of quality control and microstructure studies, certain drawbacks of the SLM technique as a method of fabrication for lattice structures were revealed. Presence of material imperfections like voids, porosity and small cracks in manufactured additively 316L stainless steel was stated. Furthermore, it was observed that the proposed powder bed fusion technique resulted in a deflection of lattice struts’ geometrical parameters. This problem is especially important in terms of numerical studies, where very often the geometrical models are defined as homogenous with the same geometrical properties. Conducted mechanical properties tests of 316L stainless steel manufactured additively, indicated the differences in comparison to the bulk material. Furthermore, results of compression tests carried out under quasi-static and dynamic conditions revealed the influence of inertia effect on deformation process and ability to energy absorption. The main damage mechanisms occurring during compression tests were described. This issue is critical in term of crashworthiness, development of technical solutions that allow a high level of safety to be ensured. 

## 2. Lattice Structure Design Process

The first stage of the conducted investigations was related to the development of 3D CAD models of lattice structure specimens. It was initially assumed that the structure samples would be defined based on body-centred cubic (BCC) topology, which can be described by four geometrical parameters: the lattice truss diameter (D), the angle between truss elements (α), the elementary unit cell height (H) and the cross-sectional shape. The aim of this investigation was to define the relationship between the specimen relative density value and energy absorption capacity. For this reason, the authors decided that the lattice specimens analyzed in this work would be described only by the lattice strut cross-section diameter (D) and the elementary unit cell height (H) (Figure 2). The lattice strut diameters (D) were defined as 0.6 mm, 0.8 mm and 1.0 mm, whereas the unit cell heights (H) were defined as 3 mm, 4 mm and 5 mm. The angle between struts (α) was the same in all cases, and its value was 70.53°. The geometrical characteristics of the designed lattice structure models are presented in Table 1.

The adopted geometrical parameters enabled obtaining specimens with similar dimensions but varied values of relative density. A detailed map of the specimen relative density in terms of the geometrical parameters is presented in Figure 3.

## 3. Lattice Structure Manufacturing Process

The complex geometry of lattice structures allows only a limited number of manufacturing methods to be implemented to build them. Considering the possibilities offered by additive manufacturing techniques, the authors decided to use an SLM device and SS316L stainless steel metal powder to produce the lattice structure specimens. Three-dimensional printing is well known and has been characterized in detail in many scientific papers [9,61,62,63,64,65]. Moreover, its growing importance from year to year can be observed. It has been applied in the production of new, cutting-edge mechanical parts used in many branches of industry. Additionally, SLM has been used in the production of ultra-light medical implants and prostheses. One of the main advantages of the SLM system is its design freedom. The adopted layer-by-layer method of object building and the use of a powder bed allows the manufacturing of hanging structures with a minimum amount of support material. This feature is particularly important in the production of objects based on 3D lattice structures. The other advantage of SLM systems is the application of a high-powered laser beam, which allows the melting of various metal powders with different mechanical properties. The designed lattice structures were manufactured using an SLM device from the Beijing Company of type AFS-M260. A standard scanning approach was applied, in which 5 × 5 mm2 islands were scanned in a random sequence. Laser power was defined as 150 W and the beam diameter was approximately 50 µm. The adopted layer thickness for SS316L fabrication was 30 µm, and the scanning velocity was approximately 1000 mm/s. All structures were produced in an argon atmosphere with a pressure of 1000–2000 mbar. Adopted technological process parameters were selected based on the results of the literature studies [64,65,66,67,68]. Additionally, specimens manufacturing process was proceeded by own technological studies realized to obtain the highest density as well as low concentration of microstructure and surface defects. These problems are typical for metal additive manufacturing and many efforts are made to solve them. Detailed information regarding the dependence between technological process parameters and material microstructure are presented in [69,70]. The SS316L powder employed (Material Technology Innovations C., Ltd., MTI S01, Guangzhou, China) was characterized by a spherical shape with a particle size in the range of 13 – 53 µm, which was in accordance with SLM requirements. The process of producing the structures was initiated by slicing the 3D CAD model into layers in order to create an *.stl file containing a 2D profiles of each layer by Materialise Magics software. Then, the *.stl file was loaded into a file preparation software package that assigned parameters, values and physical supports.

On the basis of the proposed SLM 3D printing technique, it was possible to manufacture six samples of each lattice structure variants as defined in Table 1. The main view of the specimen after cleaning is presented in Figure 4a. Additionally, Figure 4b presents a detailed view of individual elementary unit cells.

## 4. Quality Control and Microstructure Studies

The subsequent stages of the conducted investigations were associated with quality control of the lattice structure specimens. Geometrical measurements enabled the definition of dimensional deviations between the real and assumed specimen dimensions. Additionally, the smallest versions of the lattice structure specimens (with reduced global dimensions of 10 × 10 × 10 mm) were used to evaluate the material structure properties through the use of a non-destructive computer tomography technique. Applied scanning electron microscopy (SEM) as well as X-ray diffraction (XRD) tests provided the authors with a detailed evaluation of the material microstructure analysis. Due to the fact that counts of peaks intensity on the XRD pattern may differ according to various factors, e.g., different machines, other parameters during measurement, the relative intensity of peaks considering the major peak as 100% was calculated. Phase analyses were executed by XRD (Rigaku ULTIMA IV diffractometer, Rigaku Corporation, Tokyo, Japan) using Co Kα radiation for an angle range (2θ) of 40–130º with a step of 0.02º and scan speed of 1º per min. The acquired data are refined using the DHN PDF 4 crystallographic database and PDXL software (version 2.8.4.0, Rigaku Corporation, Tokyo, Japan). 

### 4.1. Geometrical Quality Control of the Lattice Structure Specimens

The first stage of the quality control process was related to an estimation of the strut diameter deviation. A geometrical evaluation was performed using the normal and smallest versions of the lattice structure specimens built via the same manufacturing process used for the specimens dedicated to the compression tests. A Keyence VHX6000 digital microscope (Keyence International, Osaka, Japan) equipped with objectives of different magnification was used to capture photographs and make measurements in dedicated software. Figure 5 presents the top and right-side views of the lattice structure specimen photographs registered for different values of the strut diameter and unit cell size. The top view presents the shape of the sample oriented in parallel in relation to the specimen building direction (layer plane) during the SLM manufacturing process.

On the basis of the analyses of captured photograph, a visible difference in the geometrical deviation of the lattice strut diameter values was observed, depending on the specimen orientation in relation to the building direction (layer plane). The chart in Figure 6 presents the assumed and real values of the lattice struts and unit cells dimension defined during measurements which were defined during measurements in the top plane (parallel to layers plane). It can be observed that the average dimensional deviation of the lattice strut diameter was approximately 0.05 mm. These divergences were mainly caused by the additive characteristics of the manufacturing process and the size of the adopted SS316L stainless steel metal powder grains. A selected fabrication layer thickness equal to 30 μm influences the surface roughness and causes the edges of material layers to be significantly visible. Further analysis of the lattice structure dimensional deviations revealed that the lattice strut diameters measured in the side-view orientation were significantly different from those measured from the top-side view. Detailed values of the dimensional deviations determined for specimens with a unit size equal to 4 mm are presented in Figure 7 and Figure 8. On the basis of the presented photographs, it can be stated that irrespective of the designed strut diameter value, the deviations range from 0 to 50 μm. Nevertheless, the dimensional deviations defined in the side-view orientations are relatively higher than those in the top-view orientation which corresponds to the building direction.

### 4.2. Structure and Microstructure Analysis of Additively Manufactured 316L Stainless Steel

Structural imperfections such as pores, voids, and microcracks were evaluated by a Nikon Metrology XTH 225 CT (Nikon Corporation, Tokyo, Japan) and a Phenom ProX SEM (Thermo Fisher Scientific Inc., Eindhoven, The Netherlands). On the basis of the captured CT images, it was possible to determine the presence of structural imperfections that could exist in additively manufactured specimens, such as small areas of porosity and voids. These problems of homogeneity in additively manufactured materials have also been observed by other scientists [69,70,71,72,73]. One of the commonly available methods for reducing the influence of structural imperfections in additively manufactured metal materials is additional postprocessing such as heat treatment or HIP (high isostatic pressure) [74,75,76]. In the present work, the sizes of the observed voids were not significant, so additional postprocessing was not applied. The measured value of the material porosity was lower than 0.5%. Figure 9presents a sampling of the captured CT images of the respective specimens. Voids marked in red color were located in various regions of the lattice specimens. Additionally, CT analysis was conducted using cylindrical specimens dedicated to the mechanical characterization of 316L stainless steel manufactured additively. On the basis of the captured CT images (Figure 10), similar material imperfections were also found. This result means that these imperfections were not related to the geometrical complexity of the fabricated objects but were generally caused by the adopted method of manufacturing. Considering the high ductility of SS316L stainless steel and the deformation of the specimens during compression, the authors determined that additional postprocessing was unnecessary.

### 4.3. Microstructural Characterization

A further material investigation was related to the microstructural analysis of 316L stainless steel. Figure 11 shows the microstructures of both the SS316L reference bulk and the SLM-produced SS316L. Images of the reference sample confirm equiaxed austenitic grains, indicating deformation and annealing twins inside the specimen, as has been proven by many authors [77,78], whereas the 316L SLM-produced samples exhibited a typical laser beam-melted morphology. The microstructural analysis performed on etched cross-sections by OM and SEM revealed the presence of hierarchical macro-, micro-, and nanostructures. Such unique microstructures arise because the SLM process is performed far from equilibrium and occurs at very high temperatures, and hence, the heating and cooling rates are high. While scanning, powder particles at the focal spot are heated to slightly above their melting point, and the shape of the melt pool formed is in accordance with the laser scanning track. Adjacent melt pools solidify rapidly, leading to densification. Inside the solidified melt pools, columnar grains elongate along the direction of thermocapillary convection of the melting tracks and heat dissipation. The aforementioned coarse grains arise because various crystallization processes can occur inside each melt pool [55]. The higher-magnification SEM images demonstrate a structure consisting of fine equiaxed grains that are elongated (some of the sub-grains in the picture are only cross-sections of individual elongated grains). Furthermore, it is worth noting that in back-scattered electron (BSE) mode, the sub-grain boundaries appeared as bright areas. According to many references [79,80,81], these boundaries are enriched in molybdenum. Saedi et al. [80,81] claimed that an extremely fast solidification rate during the SLM process leads to different solidification rates even in the melt pool, as well as different chemical composition fluctuations related to the slow kinetics of diffusion of large atoms such as molybdenum.

On the cross-section view of the SLM-produced structures, some defects with different origins are visible in Figure 12. Such defects are common in SLM and may limit the application of structures produced by this method by lowering their mechanical properties [73,74,82,83]. Defects observed in fabricated structure specimens may be classified as follows: pores, incomplete fusion holes, and cracks. The detected porosity, which consisted of pores that were mostly small in size and spherical in shape, can be attributed to insufficient powder packing during SLM; thus, the gas present between the powder particles might have dissolved in the molten pool. Pores formed from dissolved gas are trapped inside the molten pool due to extremely fast cooling. Moreover, spaces between the melt pools during successive laser passes may also be a source of so-called “inter-run porosity”. Notably, unmelted particles and incomplete fusion holes, known as lack-of-fusion (LOF) defects, may also remain inside the areas of non-overlapped laser beam tracks. The reason for the observed cracks and microcracks is the large temperature gradients and high cooling rates during SLM, which cause internal thermal stresses. Additionally, stainless steel is characterized by a low thermal conductivity and a high thermal expansion coefficient, which makes it even more vulnerable to generating cracks [73,75,80].

The XRD patterns of the 316L powder precursor and SLM-produced cellular structure are shown in Figure 13. Both XRD patterns clearly reveal the presence of a single face-centered cubic (fcc) austenite phase. It should be noted that a certain broadening of the peaks is observed which is mainly caused by the formation of the refined microstructure during the SLM process. Moreover, the slight shifting of the austenite peaks into the higher angles is noticeable implying the increase of residual stresses after the SLM. It is worth adding that the intensities of the peaks were normalized so that the strongest peaks were 100. For the SLM structure, except of the strongest peak, all the other peaks have lower intensity than those obtained for powder which indicates <111> preferential crystal orientation for the SLM process.

The chemical compositions of the SS316L powder and SLM SS316L, as presented in Table 2, showed no significant differences in the overall composition after the SLM process. It is worth noting that the obtained results are in accordance with the Schaeffler diagram, which shows the presence of phases depending on the chemical composition.

## 5. Deformation of the Lattice Structure Specimens under Various Loading Conditions

The mechanical responses of the developed and manufactured lattice structure specimens were defined based on experimental tests carried out under various (quasi-static and dynamic) loading conditions. These studies were preceded by the characterization of the mechanical properties of the applied SS316L material under various strain rate loading conditions. The obtained results are presented below.

### 5.1. Characterization of the Mechanical Properties of Additively Manufactured 316L Stainless Steel

The mechanical properties of the applied SS316L material were characterized on the basis of the results obtained during uniaxial compression tests performed under quasi-static and dynamic loading conditions. It was possible to identify the mechanical properties of SS316L stainless steel described by the Johnson–Cook (J–C) constitutive relationship, as well as to evaluate the adopted technological parameters [84,85]. This model considers the separated effects of strain hardening, the strain rate (viscosity) and thermal softening. It is represented by relation (1), where $\bar{\sigma }$ is the equivalent plastic stress (MPa), $\bar{\varepsilon }$ is the equivalent plastic strain, $\dot{\bar{\varepsilon }}$ is the equivalent plastic strain rate (s^-1^), ${\dot{\bar{\varepsilon }}}_{0}$ is the reference equivalent plastic strain rate (s^-1^), *T_m_* is the melting temperature of the material (°C) and *T_room_* is the room temperature (°C). ***A*, *B*, *C***, ***n*** and ***m*** are material parameters, which are determined based on the flow stress data obtained from mechanical tests [52].
(1)$$\bar{\sigma }=\left[A+B{\bar{\varepsilon }}^{n}\right]\left[1+Cln\left(\frac{\dot{\bar{\varepsilon }}}{{\dot{\bar{\varepsilon }}}_{0}}\right)\right]\left[1-{\left(\frac{T-T_{room}}{T_{m}-T_{room}}\right)}^{m}\right]$$

The J–C constative model is well known and is commonly used in numerical studies (CAE software, e.g., Ls-Dyna, Abaqus, Ansys, Autodyn). One of its main advantages is the consideration of the material sensitivity to the strain rate and thermal effects. Furthermore, in commonly available CAE systems, there are few versions of modified J-C constitutive material models that enable the consideration of additional material failure criteria caused by damage mechanisms [86,87,88]. Nevertheless, the J–C constitutive relations also have some drawbacks, which have caused newer, more sophisticated and more accurate descriptions to be formulated (e.g., Rusinek–Klepaczko) [89,90,91,92].

The first stage of the strength experiments was related to the definition of uniaxial compression stress-strain plots under quasi-static loading conditions (strain rate –0.001 s^-1^). These tests were performed using an universal strength machine. The subsequent step of mechanical characterization was associated with high-strain rate uniaxial compression tests, which were conducted with the use of a Split Hopkinson pressure bar (SHPB) laboratory setup. All tests for the range of strain rates from 830 to 1500 s^-1^ were carried out with the use of cylindrical tube specimens with an external diameter Ø_1_= 6 mm, an internal diameter Ø_2_ = 4 mm and a length *L* = 6 mm. As a result, it was possible to identify the mechanical parameters of the J–C material model for the SS316L material. This process was performed in accordance with the methodology described in detail in other studies [83,91]. The determined material parameters are presented in Table 3. Unfortunately, due to the lack of a heating chamber in the SHPB laboratory set-up, the thermal coefficient ***m*** was not identified.

The results of the identified mechanical properties of the additively manufactured 316L material are presented in Table 3 (first row). Comparing them to the bulk material properties and to data identified by other scientists reveals significant differences. These differences can be justified according to the adopted technological parameters used during the material manufacturing process. These conclusions are similar to those of the microstructure analyses, as described in the previous paragraphs.

### 5.2. Mechanical Response of Lattice Structure Specimens under Quasi-Static Loading Conditions

The investigations of the deformation process of the manufactured lattice structure specimens were initiated with quasi-static uniaxial compression tests. These tests were carried out with the use of a standard universal tensile test machine MTS Criterion C45 (MTS System Corporation, Eden Prairie, MN, USA) and TW-Elite software (MTS System Corporation, Eden Prairie, MN, USA), which recorded the history of the deformation process. The deformation velocity was 1 mm/s. As a result, plots of the compression process were determined for all variants of the specimen. They are presented in charts (Figure 14, Figure 15 and Figure 16) as deformation force and deformation energy curves. Additionally, the photographs in the plots below (Figure 17) illustrate the deformation process of specimens with elementary unit cells of 3, 4 and 5 and a lattice strut of 0.8.

Upon analyzing the history of the deformation force plots, it can be stated that in all cases, rupture damage of the lattice structure specimens did not occur. The registered plots were smooth, which suggested that buckling and bending effects were the main mechanisms of the deformation process. These effects were mainly caused by the high ductility property of the SS316L material used during the manufacturing process of the lattice structures. Similar conclusions are presented in the paper [55]. Comparing the curves obtained for particular specimens with various values of the unit cell size and strut diameter shows that specimens with higher relative density values were characterized by higher geometrical stiffness values (see Specimen 3_1, Specimen 4_1, Specimen 5_1) and lower ranges of deformation. The deformation energy values defined in these cases were higher than those of the other tested samples. This effect is generally caused by the geometrical relation between the strut length (which determines the unit cell size) and diameter. This situation was significantly observed for Specimen 3_1, in which the geometrical stiffness of the structure was the highest, and it was not possible to determine the highest densification of the structure due to the loading force limitation of the applied universal tensile test machine. Additionally, the slope of the deformation force increased very quickly in the analyzed case. The other interesting conclusion is that the specimens with the highest unit cell sizes also had the highest deformation range values. This effect could be crucial for energy absorption applications. The impulse of the deformation force is absorbed over a longer period of time, which means that these types of structural materials could be more effective under dynamic loading conditions.

### 5.3. Mechanical Response of the Lattice Structure Specimens under Dynamic Loading Conditions

The subsequent stage of the conducted investigations involved uniaxial compression tests under dynamic loading conditions. They were realized through the use of a dedicated version of a Split Hopkinson pressure bar laboratory setup with bar diameters of 40 mm and lengths of 3000 mm. The scheme and the main view of the applied setup are presented in Figure 18. 

Compression tests were executed using a direct-impact Hopkinson pressure bar. In this case, the specimen was glued pointwise on the frontal surface of the input bar (see scheme in Figure 18). Tests were carried out with an initial striker velocity changing from 10 m/s to 14 m/s. Depending on the tested specimen topology, the proper value of the impact velocity required to cause full densification of the structure was determined. Data reflecting the value of the impact force causing the deformation of specimens were defined based on information captured with a strain gauge (see Figure 18; strain gauge No. 1). The application of a high-speed camera (Phantom 12.1 Vision Research, Vision Research, Wayne, NJ, USA) allowed us to capture the motion of the marker defined on the side surface of the striker. On the basis of the captured videos, it was possible to define the displacement of the bar and the shortening of specimens. This process was realized through the use of additional dedicated software, Tema Motion 2D (Image Systems Motion Analysis, Linköping, Sweden). As a result of these studies, information regarding the mechanical responses of particular lattice structure specimens under dynamic loading conditions was determined. The results obtained are presented in Figure 19, Figure 20 and Figure 21. 

Analyzing the plots of the deformation force indicates that specimens with higher relative densities (Specimen 3_1, Specimen 4_1) had high geometrical stiffness values, which caused visible fluctuations during the dynamic compression tests. Specimens with lower relative density values (Specimen 3_0.6, Specimen 4_0.6, Specimen 5_0.6) were less vulnerable to fluctuations in the deformation force plot during the initial stage of compression. Nevertheless, in the final stage of deformation and before densification, they were subjected to bending, which caused a visible change in the plot history (Specimen 3_0.6, Specimen 4_0.6, Specimen 5_0.6).

Taking into consideration the entire deformation process for all analyzed cases, the history plots consist of three main elements: initial compression, a long plateau and final densification. This result means that the lattice structures perfectly absorbed and dissipated the dynamic impact and could be successfully applied in crashworthiness and passive protection systems. 

In Figure 22, the deformation processes of lattice structure specimens with different unit cell sizes are presented. On the basis of the captured images, it can be stated that the structures were compressed throughout the whole specimen volume, with no visible cracking damage mechanism causing sample fracturing. Additionally, a view of the lattice structures after dynamic testing is shown in Figure 23.

### 5.4. Comparison of Quasi-Static and Dynamic Results

Compression tests of the lattice structure specimens under various loading conditions allowed a comparison of the results obtained regarding the effect of the deformation rate. In Figure 24, Figure 25 and Figure 26, the presented results show the history plots of the deformation force determined for all considered cases of unit cell size. Analysis of the obtained data indicates that significant differences between the quasi-static and dynamic results were obtained for specimens with higher values of relative density. Moreover, these differences were caused by inertia effects, which were easy to observe under dynamic loading conditions. Referring to the data presented in Figure 24, it can be observed that a low value of the unit cell size, corresponding to a high value of the strut diameter, caused the range of plastic deformation to be short and the densification phenomenon to occur relatively quickly. Moreover, the high relative density caused deformation rate effects in all cases. Regarding the plots presented in Figure 25, it could be stated that a significant influence of inertia effects on the deformation history plot was observed for Specimen 4_1. Nevertheless, the geometrical ratio of the unit cell size to the strut diameter enabled obtaining a smooth history plot with a wide range of plateaus. By analyzing the plots presented in Figure 26, it is easy to observe that the specimen with the lowest relative density (Specimen 5_0.6) was not sensitive to the deformation rate. The curves of the deformation plots obtained under quasi-static and impact-loading conditions are almost the same. Nonetheless, before the final densification of the specimens, the bending damage mechanism occurred rapidly, which caused a visible decrease in the force value.

Moreover, in Figure 27, a comparison is presented between the maximum values of the deformation energy in terms of the relative density values. In the diagram illustrating the results determined under impact loading conditions, it is possible to observe that there is a linear relationship between the maximum values of the deformation energy in terms of the relative density. These studies were carried out with similar values of impact velocity; however, applying higher values of deformation velocity could produce different results. These conclusions are similar to those presented in another study [55]. Analyzing the data regarding the mechanical response of lattice structure specimens under quasi-static loading conditions indicated that there were some differences. The attempt to compress Specimen 3_1, which had the highest relative density, was halted due to the loading force limitation of the universal tensile machine. For this reason, the value of the maximum deformation energy in this case could be disputable. Through analyzing the rest of the results, it is difficult to define the relationship as performed for the impact loading conditions. Generally, this relation is caused by the occurrence of structure densification during the final stage of deformation.

## 6. Conclusions

In this work, regular lattice structures with different unit cell sizes and strut diameters were designed and manufactured using SS316L metal powder by the SLM process. The fabricated lattice structure specimens were verified in terms of geometrical quality control and manufacturability via the adopted technology. The main focus of the conducted studies was related to defining the mechanical behavior under both quasi-static and dynamic loading conditions. Based on the work carried out, the conclusions can be stated as follows:The fabrication of designed lattice structure specimens from 316L stainless steel metal powder is enabled using the SLM additive manufacturing technique. However, evaluations of the geometrical accuracies and microstructures revealed some drawbacks of the production process. Dimensional deviations of the lattice were different depending on the orientation of the struts in relation to the building direction\layer plane. The geometrical deflections of the lattice struts measured on the top plane (parallel to material deposition layers) were lower than the deviation measured on the side planes. This effect is generally caused by the additive nature of the manufacturing process. These discrepancies could potentially be reduced by optimizing process parameters and using powders with different grain size distribution.An evaluation of the material structure through the use of CT demonstrated the presence of pores and voids. Considering their size and stochastic distribution, additional postprocessing treatment was not applied. Nevertheless, these imperfections might affect the mechanical properties of the identified parameters of the J–C material model.The SLM-manufactured lattices revealed a structure composed of only an austenitic phase. Nevertheless, in the XRD patterns, a certain broadening and slight displacement of the austenite peaks were observed. These effects were connected to the residual stresses and refine microstructure induced by the SLM process. In addition, we observed the presence of hierarchical macro-, micro-, and nanostructures that arose during the SLM process, which was performed far from equilibrium, with high heating and cooling rates. exhibitedThe mechanical parameters of additively manufactured 316L material were different from the data gathered for the bulk material. For the additive method of specimen fabrication, the identified Johnson–Cook material model parameters will be used in further numerical simulations.Compression tests carried out under quasi-static conditions enabled observation that there is a relationship between the deformation plot history and the relative density. Specimens with higher values of relative density were characterized by a stretch-dominated mode; conversely, specimens with lower values of relative density had a bending-dominated mode with a long-range of plateaus.The results of dynamic compression tests enabled us to define the linear relationship between the energy absorption and the relative density. The dynamic behaviors of the lattice structures were similar. Specimens with higher values of relative density exhibited short stretch-dominated modes of deformation, and in turn, specimens with lower values of relative density showed long-plateau bending modes.Investigations on the technological process parameters enabling improvement of the geometrical quality of lattice specimens and the reduction of structural and microstructural imperfections must be continued. Additionally, numerical studies must be performed to define the mechanical responses of lattice structure specimens with a wide range of geometrical parameters. The results of the performed experimental studies will be used in the validation of the proposed numerical model.Lattice structure materials made of SS316L show promise for use as a new light constructional material with high mechanical properties. They can be potentially used in further cutting-edge products in many industrial fields.

## Figures and Tables

**Figure 1 materials-13-02204-f001:**
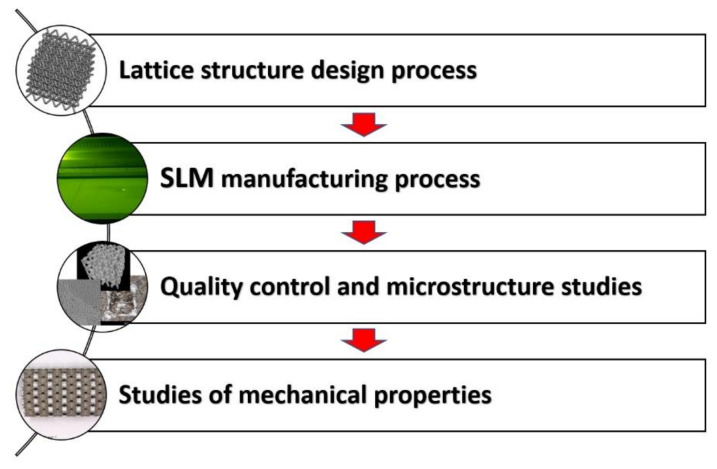
The main steps of the conducted investigation methodology.

**Figure 2 materials-13-02204-f002:**
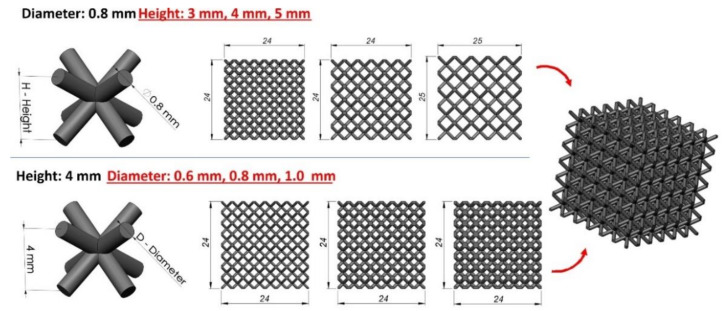
Geometrical assumptions undertook during the specimen manufacturing process.

**Figure 3 materials-13-02204-f003:**
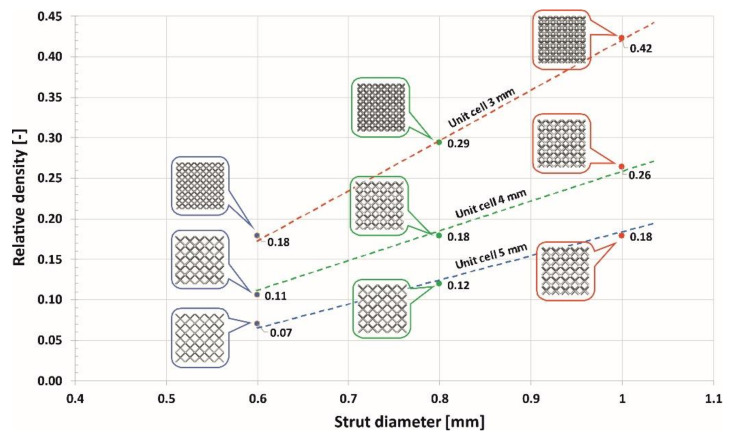
Map of the relationship between the value of the structure relative density and the adopted geometrical parameters.

**Figure 4 materials-13-02204-f004:**
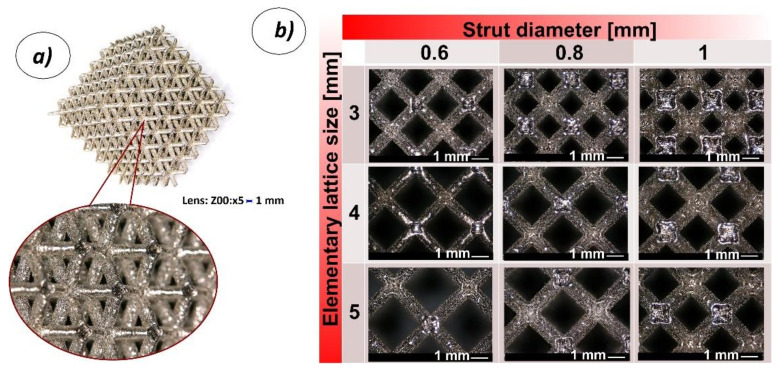
View of a lattice structure specimen made additively with the use of the selective laser melting (SLM) 3D printing technique; **a**) isometric view of the structure, **b**) detailed views of elementary unit cells (3, 4, 5 refer to the dimensions of the elementary cell size; 0.6, 0.8, 1 refer to the strut thickness).

**Figure 5 materials-13-02204-f005:**
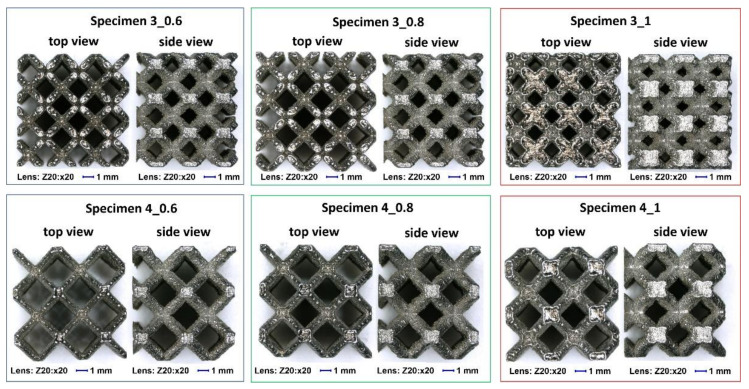
Digital microscope photographs of the top and side views of the lattice structure specimens (magnification ×20).

**Figure 6 materials-13-02204-f006:**
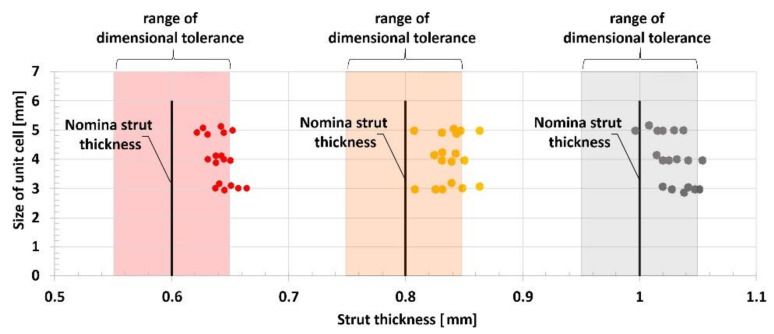
The dimensional deflections of the lattice structure struts relative to the assumed strut diameter and unit cell size.

**Figure 7 materials-13-02204-f007:**
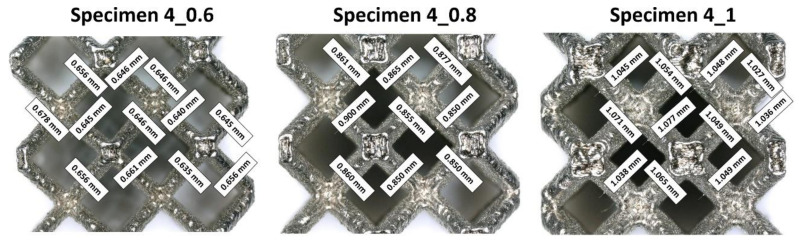
The dimensional deflections of the lattice structure struts measured from the top view (parallel to the building direction).

**Figure 8 materials-13-02204-f008:**
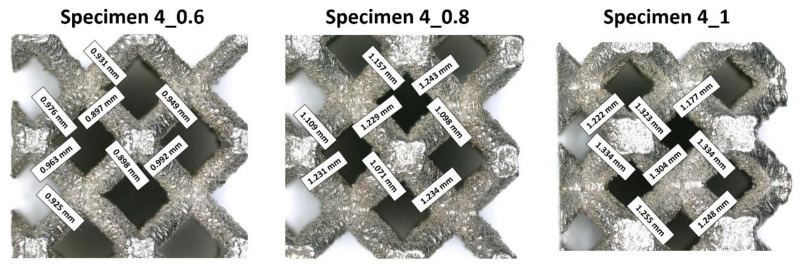
The dimensional deflections of the lattice structure struts measured from the side view (perpendicular to the building direction).

**Figure 9 materials-13-02204-f009:**
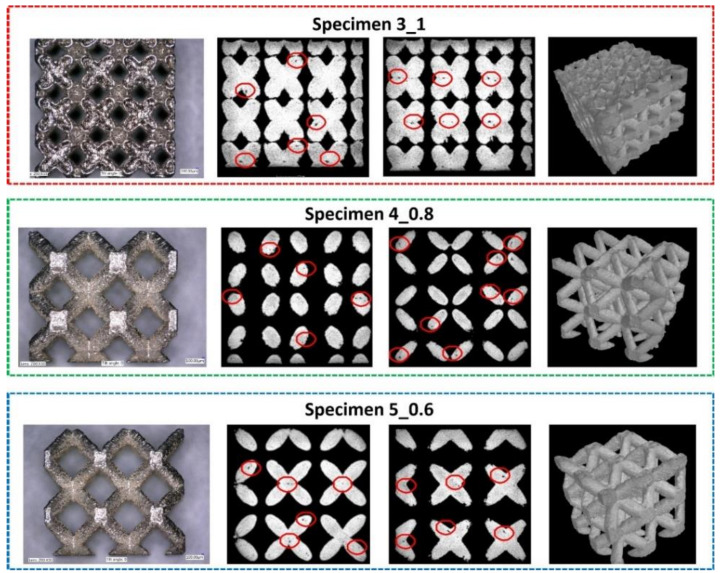
Sample images of structural analysis of lattice structure specimens with the use of CT.

**Figure 10 materials-13-02204-f010:**
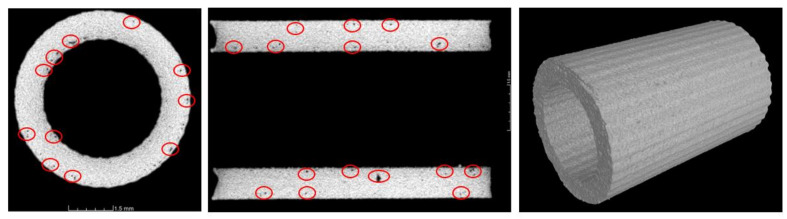
Example CT images of the cylindrical tube specimens.

**Figure 11 materials-13-02204-f011:**
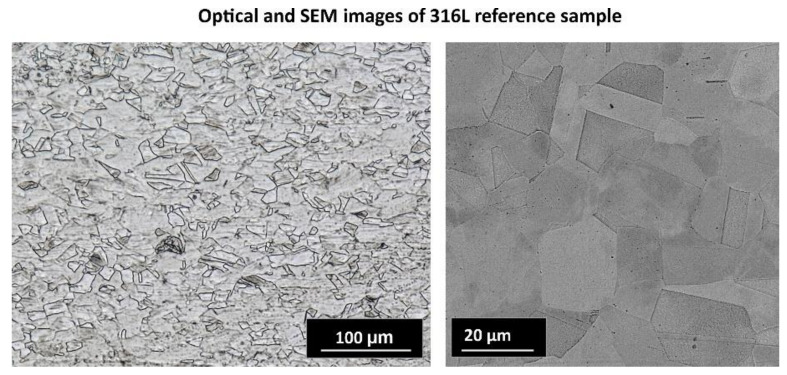
Digital and scanning electron microscope (SEM) images of the reference sample showing equiaxed austenitic grains as well as the SLM-produced structure revealing the characteristic morphology of laser-melted techniques.

**Figure 12 materials-13-02204-f012:**
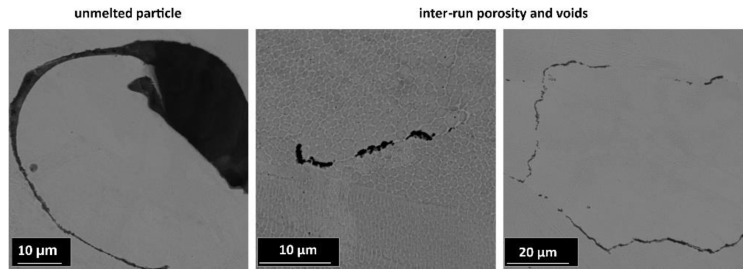
SEM images showing microstructure defects.

**Figure 13 materials-13-02204-f013:**
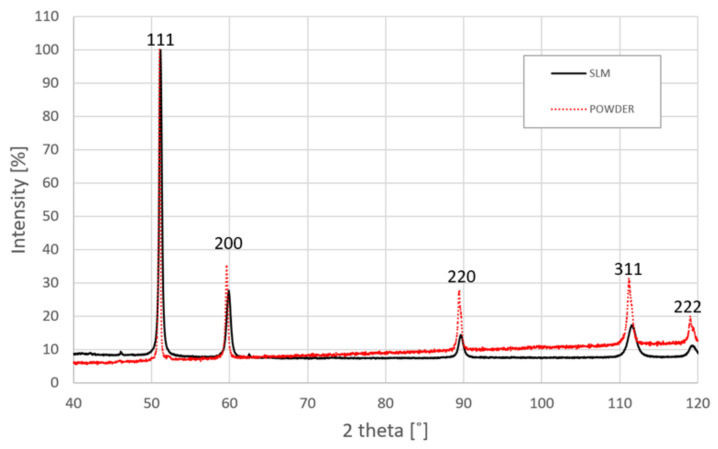
The X-ray diffraction (XRD) pattern of the SS316L structure produced by SLM.

**Figure 14 materials-13-02204-f014:**
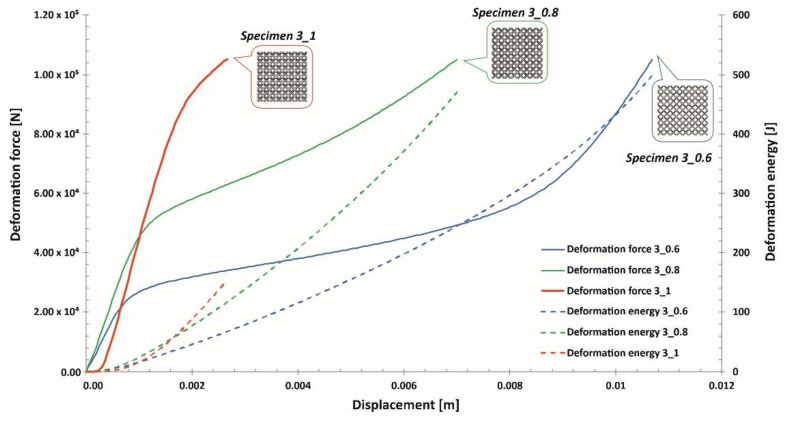
Deformation force and energy plots determined for lattice structures with an elementary unit cell equal to 3 mm.

**Figure 15 materials-13-02204-f015:**
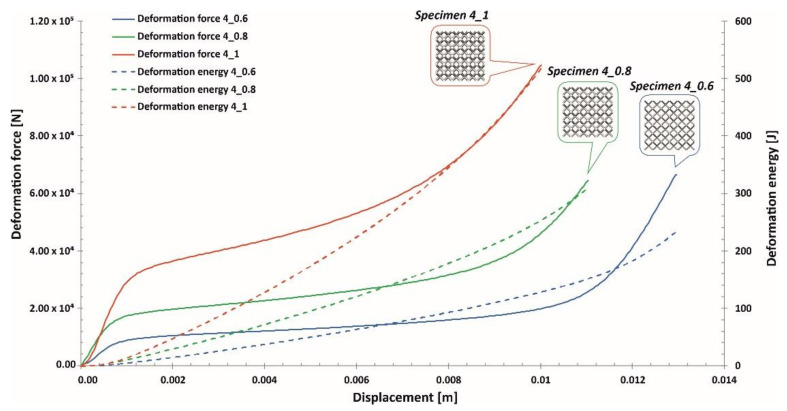
Deformation force and energy plots determined for lattice structures with an elementary unit cell equal to 4 mm.

**Figure 16 materials-13-02204-f016:**
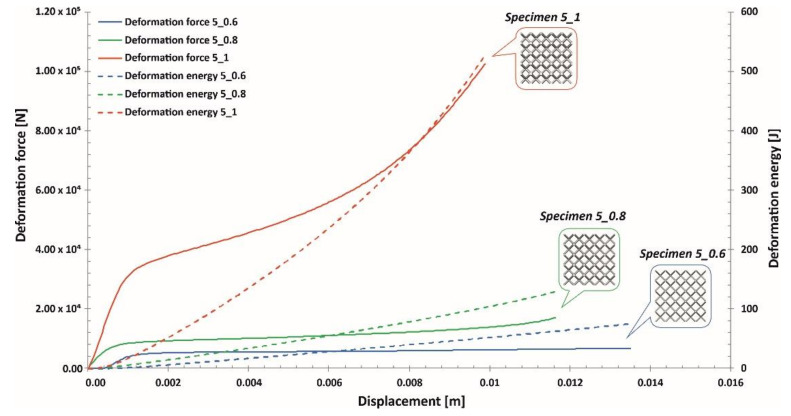
Deformation force and energy plots determined for lattice structures with an elementary unit cell equal to 5 mm.

**Figure 17 materials-13-02204-f017:**
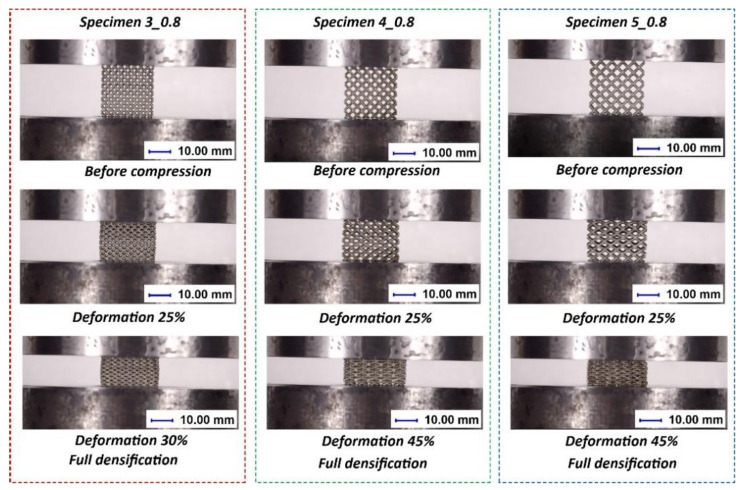
Exemplary views of the specimen deformation process.

**Figure 18 materials-13-02204-f018:**
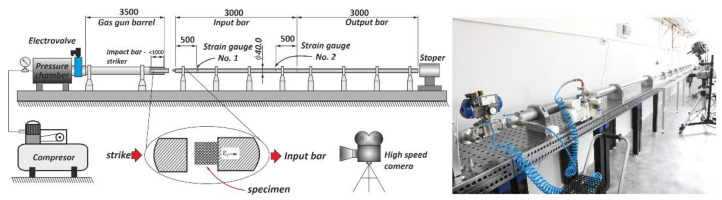
Scheme and main view of the Split Hopkinson pressure bar laboratory setup with the indirect configuration used to perform dynamic compression tests.

**Figure 19 materials-13-02204-f019:**
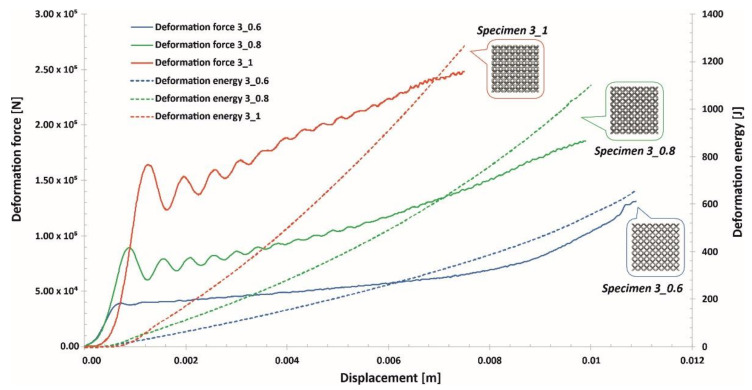
Deformation force and energy plots determined for lattice structures with an elementary unit cell equal to 3 mm under dynamic tests.

**Figure 20 materials-13-02204-f020:**
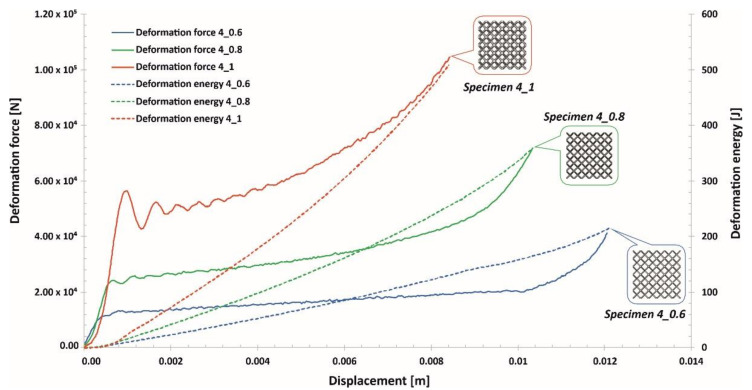
Deformation force and energy plots determined for lattice structures with an elementary unit cell equal to 4 mm under dynamic tests.

**Figure 21 materials-13-02204-f021:**
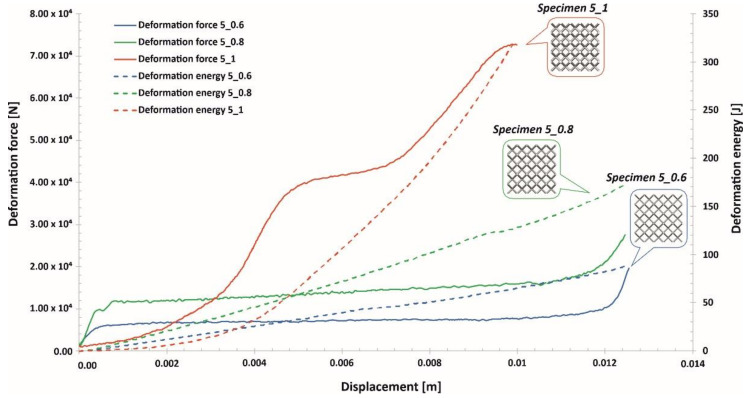
Deformation force and energy plots determined for lattice structures with an elementary unit cell equal to 5 mm under dynamic tests.

**Figure 22 materials-13-02204-f022:**
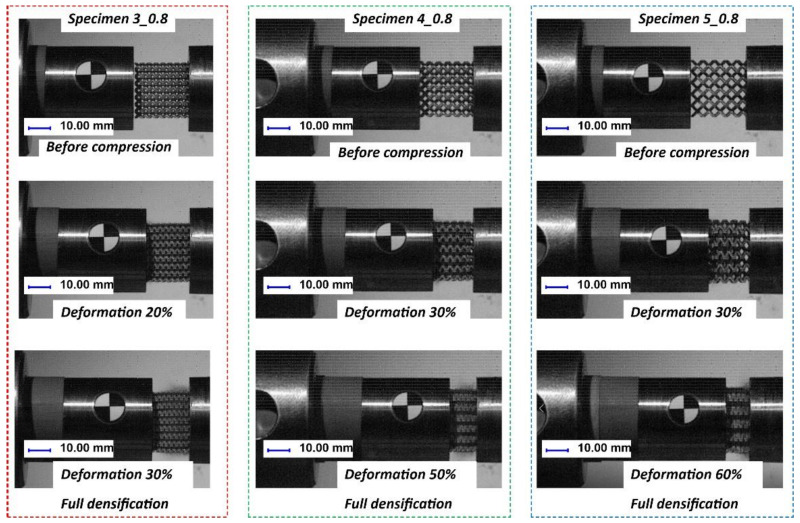
Example views of the deformation processes of lattice structures with different unit cell size values under impact loading conditions.

**Figure 23 materials-13-02204-f023:**
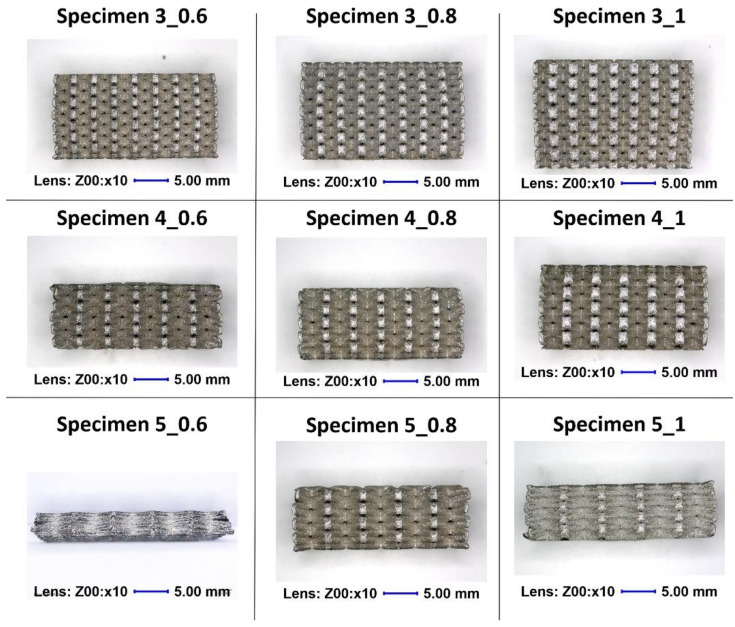
Lattice structure views after impact testing.

**Figure 24 materials-13-02204-f024:**
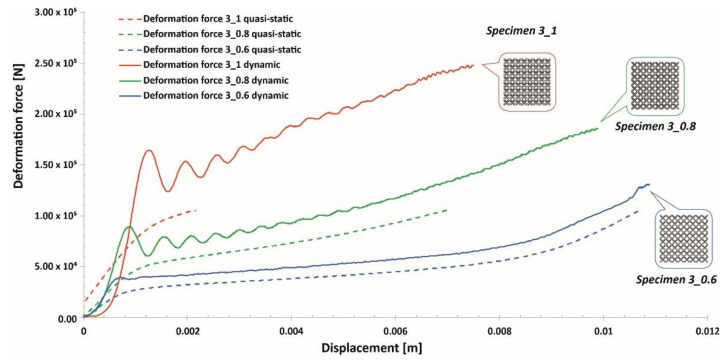
Comparison of the deformation force plots of lattice structure specimens with a 3 mm unit cell size under quasi-static and dynamic loading conditions.

**Figure 25 materials-13-02204-f025:**
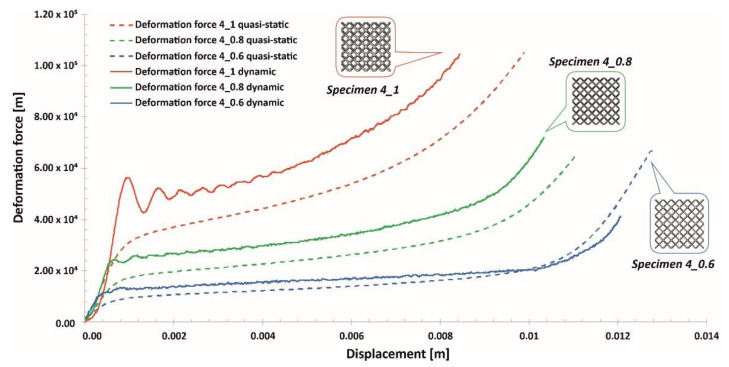
Comparison of the deformation force plots of lattice structure specimens with a 4 mm unit cell size under quasi-static and dynamic loading conditions.

**Figure 26 materials-13-02204-f026:**
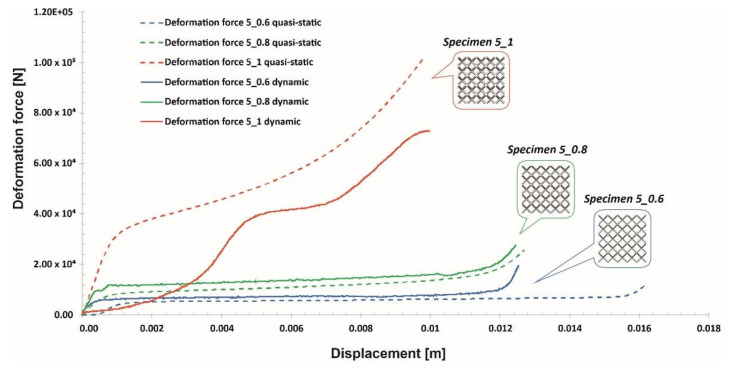
Comparison of the deformation force plots of lattice structure specimens with a 5 mm unit cell size under quasi-static and dynamic loading conditions.

**Figure 27 materials-13-02204-f027:**
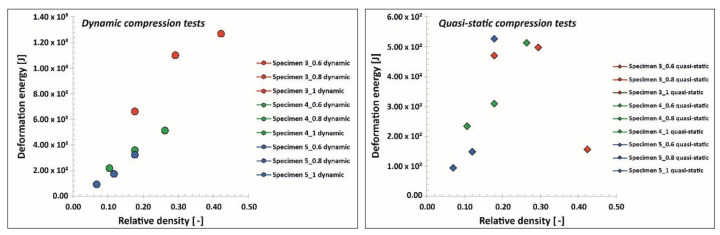
Comparison of the maximum deformation energy values under quasi-static and impact loading conditions.

**Table 1 materials-13-02204-t001:** Geometrical specifications of the developed lattice structures.

No.	Strut Diameter D [mm]	Height of Unit Cell H [mm]	Dimensions of Specimen [mm]	Mass [g]	Relative Density ρ_rel_ [-]
Specimen 3_0.6	0.6	3	24 × 24 × 24	19.73	0.18
Specimen 3_0.8	0.8	3	24 × 24 × 24	32.51	0.29
Specimen 3_1	1	3	24 × 24 × 24	46.79	0.42
Specimen 4_0.6	0.6	4	24 × 24 × 24	11.75	0.11
Specimen 4_0.8	0.8	4	24 × 24 × 24	19.73	0.18
Specimen 4_1	1	4	24 × 24 × 24	29.14	0.26
Specimen 5_0.6	0.6	5	25 × 25 × 25	8.73	0.07
Specimen 5_0.8	0.8	5	25 × 25 × 25	14.90	0.12
Specimen 5_1	1	5	25 × 25 × 25	22.30	0.18

**Table 2 materials-13-02204-t002:** Chemical compositions of the SS316L powder and SLM 316L (all in wt %).

Element	C	Cr	Ni	Mn	Mo	Fe
SS316L Powder	0.014	17.5	11.5	<2	2.3	Balance
SLM SS316L	-	16.89	11.54	1.35	2.78	Balance

**Table 3 materials-13-02204-t003:** SS316L material parameters identified through the use of Johnson–Cook constitutive relations.

No.	A [MPa]	B [MPa]	n [-]	C [-]	m [-]
316L manufactured additively [authors]	542	303	0.293	0.028	-
SS 316L bulk [authors]	304	1097	0.492	0.014	-
316L manufactured additively [93]	380	825	0.726	0.115	-
316L manufactured additively [94]	280	767	0.587	-	-
316L manufactured additively [95]	310.8	881.38	0.178	0.19	-

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
