# Peer review of "Investigations on Mechanical Properties of Lattice Structures with Different Values of Relative Density Made from 316L by Selective Laser Melting (SLM)"

_materials, 2020, doi:10.3390/ma13092204_

Round 1
Reviewer 1 Report
General comments
Paper is quite well structured and well referenced. English is also quite good.
Results are reasonably new and have a good potential for future applications.
There are certain comments and suggestions but nothing related to the content and value of the presented results.
Particular comments
Comments that need certain explanations are listed below. Suggestion on the minor corrections not listed are shown in the attached scan of the paper with handwritten remarks.
(o)!!
There is a number of publications related to the studies into the mechanical properties of 316L lattices made by different powder bed fusion technologies. So in the Introduction it would be advisable to have a comment on what is specifically different in particular research (for example, comparison of static and dynamic loading). In such case, reader will understand that other results {the deflections of the strut dimensions from what is demanded by the design files, the difference of the microstructure of the PBF-manufactured material from the microstructure of 316L manufactured by other methods, and the differences in the microstructure of the thin struts as compared to ‘bulk’ ones from the same PBF method} are supportive for the main results (comparison of static and dynamic mechanical loading and energy absorption). If not done so, the question is arising as many of the reported results are seemingly repeating what have been done by others and in more thorough way.
Additionally it should be clearly stated, that present paper is specifically addressing the results for SLM-manufactured regular lattices, which will approve the lack of references to the results coming from EBM side.
!!
(i) Abstract, 4th line from top:
‘ These deviations depend on the direction of geometrical evaluation’
Term ‘direction of geometrical evaluation’ is unclear. Similar issues one can see in some places of the text -see pos. (ii) and (iii) below.
Please use better definition, especially in the Abstract where there is no possibility of adding explanatory text.
In the abstract one can possibly use: ‘These deviations depend on the strut orientation in relation to the build direction’: properties do not depend on the plane you choose to study, but on the direction of the struts in the build
(ii) page 6, lines 186-200
Similar to (i) above, it is not direction of the measurements which matters, it is position of the struts in the sample (top layers, bottom layers, sides of the sample) and direction of the struts in relation to the build direction.
To avoid misunderstanding you need to define:
- the shape of the samples, orientation of the samples in relation to the build direction (layer plane);
- what is defining the difference in strut dimensions- strut position in the sample (sides, top or bottom- if the sample cubes are oriented with the ‘top’ and ‘bottom’ are parallel to the layer plane), or direction of the struts in relation to the build direction
Similar issues should be clarified through the text (for example in the captures to Figures 7 and 8)
(iii) Conclusions, (I)
Same problems, as described in (i) and (ii)
‘… depending on the direction of geometrical evaluation’
(iv) Figure 4 (b)
Numbers given as references (0.6, 0.8, 1 and 3, 4, 5) should be briefly described in the figure capture, something like :’b) detailed views of elementary unit cells (3, 4 , 5 refer to the dimensions of the elementary cell size; 0.6, 0.8, 1 refer to the strut thickness- as demanded by the build files).’
(v) Additionally to (iv)
You should clearly define the references you are using in the text referring to particular samples- like ‘specimen 3_0.8’ so the reader will further understand what do the numbers mean.
Probably that shouold be added in ‘2 Lattice design process’
(vi) page 9, line 252-254- possibly a missing word?
‘extremely fast solidification rate during the SLM process leads to different solidification rates even in the melt pool, as well as chemical composition fluctuations related to the slow kinetics of large atoms such as molybdenum.’
‘the slow kinetics ‘ term kinetics commonly needs some clarification- kinetics of diffusion, kinetics of decay etc.-e.g. ‘kinetics of (process) of large atoms’?
(vii) page 10, lines 281-282
‘…After the SLM process, no noticeable crystallographic orientation was observed….’
Probably- missing words? Possibly:
‘…After the SLM process, no noticeable CHANGES IN crystallographic orientation was observed….’
(viii) Figures 14-16, 19-21and 24-26
Number of figures seems little excessive. Figures partially show same things. Moreover, main extracted parameters are further presented in the discussion. It is suggested to use one of three Figures in each series as an example going down from 9 figures to 3.
If you have doubts, consult with the Journal Editorial board what would they suggest in this respect.
(ix) Figure 23, left bottom corner panel (Specimen 5_06)
Direction of the compressed sample is 90 degrees rotated as compared to the other panels, and extra elements are visible.
It is better to present all samples in a similar way!
(x) Conclusions (I), last sentence:
‘…These discrepancies could potentially be reduced by applying a thinner manufacturing layer….’
It is at least very weak or even doubtful statement. Many of the problems with the differences of the actual strut thickness in as-manufactured lattice struts in PBF depend on the spreading of the melt pool beyond the beam ‘spot’ dimensions. Powder grain size and layer thickness is only in part influencing such spread. Process parameters (energy deposition rate, scanning strategy etc.) also have quite serious impact.
It is better to use softer formulation, something like:
‘‘…These discrepancies could potentially be reduced by optimizing process parameters and using powders with different grain size distribution….’

Author Response
Dear Reviewer,
At the beginning, the Authors would like to express their great thanks for the effort in a very thorough analysis of the article in terms of its scientific aspects and style. The comments are valuable and helpful for revising and improving the paper. The comments have been studied carefully, and the correction has been introduced.
Particular comments
Comments that need certain explanations are listed below. Suggestions on the minor corrections not listed are shown in the attached scan of the paper with handwritten remarks.
Comment 1)
There is a number of publications related to the studies into the mechanical properties of 316L lattices made by different powder bed fusion technologies. So in the Introduction it would be advisable to have a comment on what is specifically different in particular research (for example, comparison of static and dynamic loading). In such case, reader will understand that other results {the deflections of the strut dimensions from what is demanded by the design files, the difference of the microstructure of the PBF-manufactured material from the microstructure of 316L manufactured by other methods, and the differences in the microstructure of the thin struts as compared to ‘bulk’ ones from the same PBF method} are supportive for the main results (comparison of static and dynamic mechanical loading and energy absorption). If not done so, the question is arising as many of the reported results are seemingly repeating what have been done by others and in more thorough way.
Additionally it should be clearly stated, that present paper is specifically addressing the results for SLM-manufactured regular lattices, which will approve the lack of references to the results coming from EBM side.
Response:
Thank you very much for your valuable comment. The last section of the introduction was modified to underline the main goal of the presented paper in a way that was suggested by the Reviewer.
“The main goal of the paper is a presentation of results of experimental investigations related to the mechanical properties of lattice structures made additively from SS316L stainless steel. Developed specimens with different values of the relative density were manufactured via selective laser sintering SLM and submitted to mechanical and structural tests according to the methodology presented in Figure 1. This paper is a continuation of Authors’ works related to additively manufactured lattice structure made from 316L stainless steel. Results associated with this work were partially published in [62,63]. On the basis of the results of quality control and microstructure studies, it was revealed certain drawbacks of the SLM technique as a method of fabrication of the lattice structures. Presence of material imperfections like voids, porosity, and small cracks in manufactured additively 316L stainless steel was stated. Furthermore, it was observed that the proposed powder bed fusion technique resulted in a deflection of lattice struts geometrical parameters. This problem is especially important in terms of numerical studies, where very often the geometrical models are defined as a homogenous with the same geometrical properties. Conducted mechanical properties tests of 316L stainless steel manufactured additively, indicated the differences in comparison to the bulk material. Furthermore, results of the compression tests carried out under quasi-static and dynamic conditions revealed the influence of inertia effect on deformation process and ability to energy absorption. The main damage mechanisms arriving during compression test were described. This issue is critical in terms of crashworthiness, development of technical solutions that allow ensuring a high level of safety. “
Comment 2)
(i) Abstract, 4th line from top:
‘ These deviations depend on the direction of geometrical evaluation’
Term ‘direction of geometrical evaluation’ is unclear. Similar issues one can see in some places of the text -see pos. (ii) and (iii) below.
Please use better definition, especially in the Abstract where there is no possibility of adding explanatory text.
In the abstract one can possibly use: ‘These deviations depend on the strut orientation in relation to the build direction’: properties do not depend on the plane you choose to study, but on the direction of the struts in the build
Thank you for your comment. Of course, you are right, the value of strut diameter does not depend on the direction of measurements (observation) but it depends on the specimen building direction. These discrepancies may cause a serious problem especially in numerical investigations of specimens deformation process, where very often the numerical model is defined with assumption that strut shape is perfectly circular.
The following change was proposed:
“These deviations depend on the strut orientation in relation to the specimen building direction”.
Comment 3)
(ii) page 6, lines 186-200
Similar to (i) above, it is not direction of the measurements which matters, it is position of the struts in the sample (top layers, bottom layers, sides of the sample) and direction of the struts in relation to the build direction.
Response: Thank you for your comment. Of course, it sounds better if the following information will be added on page 6:
„ On the basis of the analyses of captured photographs, a visible difference in the geometrical deviation of the lattice strut diameter values was observed, depending on the specimen orientation in relation to the building direction (layer plane). The chart in Figure 6 presents the assumed and real values of the lattice struts and unit cell dimensions which were defined during measurements in the top plane (parallel to layers plane)”.
Comment 4)
To avoid misunderstanding you need to define:
- the shape of the samples, the orientation of the samples in relation to the build direction (layer plane);
- what is defining the difference in strut dimensions- strut position in the sample (sides, top or bottom- if the sample cubes are oriented with the ‘top’ and ‘bottom’ are parallel to the layer plane), or direction of the struts in relation to the build direction
Similar issues should be clarified through the text (for example in the captures to Figures 7 and 8)
Resposne: Thank you for your remark. An appropriate modification was introduced in the mentioned section of the chapter. Our intention was a presentation of the problem of geometrical deviation in relation to the building direction.
The following captions were proposed below Figure. 7 and Figure. 8.
Figure 7. The dimensional deflections of the lattice structure struts measured from the top view (parallelly to the building direction)
Figure 8. The dimensional deflections of the lattice structure struts measured from the side view (perpendicular to the building direction)
Comment 5)
(iii) Conclusions, (I)
Same problems, as described in (i) and (ii)
‘… depending on the direction of geometrical evaluation’
Response: Thank you for your comment. This sentence was modified.
Comment 6)
(iv) Figure 4 (b)
Numbers given as references (0.6, 0.8, 1 and 3, 4, 5) should be briefly described in the figure capture, something like :’b) detailed views of elementary unit cells (3, 4 , 5 refer to the dimensions of the elementary cell size; 0.6, 0.8, 1 refer to the strut thickness- as demanded by the build files).’
Response: Thank you for your comment. Additional information relating to elementary size and strut diameter were placed in Fig.4 and the caption was modified according to the Reviewer suggestion
Comment 7)
(v) Additionally to (iv)
You should clearly define the references you are using in the text referring to particular samples- like ‘specimen 3_0.8’ so the reader will further understand what do the numbers mean.
Probably that should be added in ‘2 Lattice design process’
Response: Thank you for your comment. This information is placed at the beginning of chapter 2. The detailed information regarding dimension parameters are presented in Fig.2 and additionally mentioned in Table 1. Furthermore, an appropriate comment is added in the introduction to chapter 2.
Comment 8)
(vi) page 9, line 252-254- possibly a missing word?
‘extremely fast solidification rate during the SLM process leads to different solidification rates even in the melt pool, as well as chemical composition fluctuations related to the slow kinetics of large atoms such as molybdenum.’
‘the slow kinetics ‘ term kinetics commonly needs some clarification- kinetics of diffusion, kinetics of decay etc.-e.g. ‘kinetics of (process) of large atoms’?
Response: Thank you for your remark. This sentence was changed.
Comment 9)
‘…After the SLM process, no noticeable crystallographic orientation was observed….’
Probably- missing words? Possibly:
‘…After the SLM process, no noticeable CHANGES IN crystallographic orientation was observed….’
Response: Thank you for your remark. This sentence was changed.
Comment 10)
(viii) Figures 14-16, 19-21and 24-26
Number of figures seems little excessive. Figures partially show same things. Moreover, main extracted parameters are further presented in the discussion. It is suggested to use one of three Figures in each series as an example going down from 9 figures to 3.
If you have doubts, consult with the Journal Editorial board what would they suggest in this respect.
Response: Figures from 14 to 16 present the results of compression tests under quasi-static loading conditions. Figures from 19 to 21 present the results registered under dynamic compression tests. Figures from 24- to 26 presents the comparison between quasi-static and dynamic compression. Each figure demonstrates history plots (deformation force and deformation energy ) registered for the other group of structures variants. In the Authors' opinion separation of results depending on the size of the elementary unit cell enables better analysis of these results. Furthermore, figures from 24 to 26 demonstrate the influence of the inertia effect and its influence on the ability to energy absorption. If it is not so serious problem we would like to presents results in the proposed form.
Comment 11)
(ix) Figure 23, left bottom corner panel (Specimen 5_06)
Direction of the compressed sample is 90 degrees rotated as compared to the other panels, and extra elements are visible.
It is better to present all samples in a similar way!
Response: Thank you for your remark. Figure 23 was changed according to the Reviewer suggestion. It looks better.
Comment 12)
(x) Conclusions (I), last sentence:
‘…These discrepancies could potentially be reduced by applying a thinner manufacturing layer….’
It is at least very weak or even doubtful statement. Many of the problems with the differences of the actual strut thickness in as-manufactured lattice struts in PBF depend on the spreading of the melt pool beyond the beam ‘spot’ dimensions. Powder grain size and layer thickness is only in part influencing such spread. Process parameters (energy deposition rate, scanning strategy etc.) also have quite serious impact.
It is better to use softer formulation, something like:
‘‘…These discrepancies could potentially be reduced by optimizing process parameters and using powders with different grain size distribution….’
Response: Thank you for your remark. The mentioned sentence was changed according to the Reviewer suggestion. It sounds better.
“These discrepancies could potentially be reduced by optimizing process parameters and using powders with different grain size distribution.”
Reviewer 2 Report
The topic is quite interesting, however the manuscript has some relevant flaws listed below. The publication could be considered after major revisions.
- It was produced a single sample for each geometry. Information about the reproducibility should be given. In my opinion this is a serious lack in the soundness of results and conclusions.
- The authors have to indicate what represent the points reported in Figure 6.
- Row 218. “In the present work, the sizes of the observed voids were not significant, so additional postprocessing was not applied”. The authors should explain what they mean “significant” and under which conditions postprocessing must be made.
- Comments to Figure 13 (XRD). This part is very weak and must be completely rewritten.
- The authors must indicate the experimental conditions (radiation used in present measurements, angular step, counting time per step etc.).
- Are the XRD patterns the same in all the directions of SLM samples?
- Row 279: “It should be noted that a certain broadening of the peaks is observed due to residual stresses and related lattice distortion induced by the SLM process”. The sentence is confused and misleading. The main effect of residual stresses (σI stresses) is to shift the XRD peaks; by determining with high precision the shift (it is not the case of the pattern reported in Figure 6) it is possible to determine the residual stresses. Broadening is mainly due to grain size, and to σII and σIII stresses.
- Row 280: “Moreover, the displacement of the austenite peaks is caused by the highly directional growth and the presence of texture. After the SLM process, no noticeable crystallographic orientation was observed.“ The displacement of austenitic peaks is not due to the texture, which instead affects the relative intensities of peaks. In figure 13 the intensities of the strongest peaks have been normalized to 100; all the other peaks are of lower intensity in SLM samples with respect those obtained from the powder thus SLM gives rise to a stronger {111}orientation. Therefore, the comment “no noticeable crystallographic orientation was observed“ is not correct.
Author Response
Dear Reviewer,
At the beginning, the Authors would like to express their great thanks for the effort in a very thorough analysis of the article in terms of its scientific aspects. The comments are valuable and helpful for revising and improving the paper. The comments have been studied carefully, and the correction has been introduced. The changes are marked in the paper.
Comment 1)
- It was produced a single sample for each geometry. Information about the reproducibility should be given. In my opinion this is a serious lack in the soundness of results and conclusions.
Response: Thank you for your comment. You are right that lack information about a number of the specimen can cause doubts about the repeatability of results. The number of one variant samples was six. Three of them were used to static and the other to dynamic tests. Obtained results of mechanical tests were similar and good repeatability of compression history plots was stated for both, quasi-static and dynamic tests. In accordance with the Reviewer’s suggestion, this information was added to chapter 2 that refers to the specimens’ manufacturing process.
“On the basis of the proposed SLM 3D printing technique, it was possible to manufacture six samples of each lattice structure variants as defined in Table 1.”
Comment 2)
The authors have to indicate what represent the points reported in Figure 6.
Response: Thank you for your remark. Points presented in Fig.6 refers to the number of structure samples which were submitted to the geometrical quality control process. Figure.6 was modified in accordance with the Reviewer comment.
Comment 3)
Row 218. “In the present work, the sizes of the observed voids were not significant, so additional postprocessing was not applied”. The authors should explain what they mean “significant” and under which conditions postprocessing must be made.
Response: Thank you for your comment. Conducted porosity measurements of 316L stainless steel manufactured additively via SLM did not exceed the value of 0.5 %. That was the reason why the Authors decided that additional postprocessing like heat treatment or HIP was not demanded. Of course, the Reviewer is right that appropriate explanation should be placed to avoid misunderstood. In our opinion, the necessity of additional postprocessing depends on the kind of applied material and geometrical features of manufactured additively objects (thick walls, large surfaces with a tendency to deformation caused by residual stress).
Comment 4)
Comments to Figure 13 (XRD). This part is very weak and must be completely rewritten.
Response: The part about XRD has been rewritten.
Comment 5)
The authors must indicate the experimental conditions (radiation used in present measurements, angular step, counting time per step etc.).
Response: Experimental conditions of XRD measurement has been added.
Comment 6)
Are the XRD patterns the same in all the directions of SLM samples?
Response: XRD pattern has been done only in one direction.
Comment 7)
Row 279: “It should be noted that a certain broadening of the peaks is observed due to residual stresses and related lattice distortion induced by the SLM process”. The sentence is confused and misleading. The main effect of residual stresses (σI stresses) is to shift the XRD peaks; by determining with high precision the shift (it is not the case of the pattern reported in Figure 6) it is possible to determine the residual stresses. Broadening is mainly due to grain size, and to σII and σIII stresses.
Response: Thank you very much for your valuable comment. Indeed, the sentence you cited may be confusing and misleading. So that the sentence was corrected.
Comment 8)
Row 280: “Moreover, the displacement of the austenite peaks is caused by the highly directional growth and the presence of texture. After the SLM process, no noticeable crystallographic orientation was observed.“ The displacement of austenitic peaks is not due to the texture, which instead affects the relative intensities of peaks. In figure 13 the intensities of the strongest peaks have been normalized to 100; all the other peaks are of lower intensity in SLM samples with respect those obtained from the powder thus SLM gives rise to a stronger {111}orientation. Therefore, the comment “no noticeable crystallographic orientation was observed“ is not correct.
Response: Thank you very much for your valuable comment. The section you mentioned was corrected according to your instructions.
Round 2
Reviewer 2 Report
After corrections the manuscript can be published.